# Breeding practices and trait preferences of smallholder farmers for indigenous sheep in the northwest highlands of Ethiopia: Inputs to design a breeding program

Abiye Shenkut Abebe[1,2]*, Kefyalew Alemayehu[1], Anna Maria Johansson[3], Solomon Gizaw[4]

1 Department of Animal Production and Technology, Bahir Dar University, Bahir Dar, Ethiopia, 2 Department of Animal Science, Debre Tabor University, Debre Tabor, Ethiopia, 3 Department of Animal Breeding and Genetics, Swedish University of Agricultural Sciences, Uppsala, Sweden, 4 International Livestock Research Institute (ILRI), Addis Ababa, Ethiopia

* abiysh85@gmail.com

**Data Availability Statement:** All relevant data are within the manuscript and its Supporting Information files.

## Abstract

The aim of this study was to identify breeding practices and trait preferences for indigenous sheep in three districts (Estie, Farta and Lay Gayient) located in the northwest highlands of Ethiopia. Questionnaire survey and choice experiment methods were used to collect data from 370 smallholder farmers. Respondents were selected randomly among smallholder farmers who own sheep in the aforementioned districts. A generalized multinomial logit model was employed to examine preferences for sheep attributes, while descriptive statistics and index values were computed to describe sheep breeding practices. Having the highest index value of 0.36, income generation was ranked as the primary reason for keeping sheep, followed by meat and manure sources. The average flock size per smallholder farmer was 10.21 sheep. The majority of the smallholder farmers (91%) have the experience of selecting breeding rams and ewes within their own flock using diverse criteria. Given the highest index value of 0.34, body size was ranked as a primary ram and ewe selection criteria, followed by coat color. Furthermore, choice modeling results revealed that tail type, body size, coat color, growth rate, horn and ear size have shown significant influences on smallholder farmers' preference for breeding rams (P<0.01). The part-worth utility coefficients were positive for all ram attributes except ear size. For breeding ewes, mothering ability, coat color, body size, lambing interval, growth rate, tail type and litter size have shown significant effects on choice preferences of smallholder farmers (P<0.05). Moreover, significant scale heterogeneity was observed among respondents for ewe attributes (P<0.001). Overall, the results implied that sheep breeding objectives suitable for the northwest highlands of the country can be derived from traits such as linear body measurement, weight and survival at different ages, and lambing intervals. However, selection decisions at the smallholder level should not only be based on estimated breeding values of traits included in the breeding objective but instead, incorporate ways to address farmers' preference for qualitative traits.

**Funding:** ASA received fund for this study from the Ethiopian Ministry of Education via College of Agriculture and Environmental Sciences of Bahir Dar University (Ref.No/250/2019). The website for College of Agriculture and Environmental Sciences of Bahir Dar University is https://bdu.edu.et/caes/. The funders had no role in study design, data collection and analysis, decision to publish, or preparation of the manuscript.

**Competing interests:** The authors have declared that no competing interests exist.

## Introduction

Ethiopia is one of the few African countries endowed with huge sheep genetic resources. The estimated population size is about 31.3 million sheep, of which 99.8% are indigenous types [1]. Ecologically, sheep are found in diverse production environments that range from arid lowlands to extremely cool highlands. The northwest highlands are among the major sheep production areas of the country, where Farta sheep breed and their crosses with other indigenous sheep are widely distributed [2, 3]. For the smallholder farmers, sheep provide valuable contributions through income generation, direct food sources, non-food utilities and various socio-cultural privileges [3, 4, 5]. Particularly when crop farming is less reliable due to drought or other factors, sheep are commonly used to mitigate adverse effects, for instance, related to food shortage at the smallholder level.

Despite the presence of a large number of sheep and their diverse functions, the average productivity of indigenous sheep is generally low. For example, Mekuriaw et al. [6] reported an average yearling weight of about 20 kg for Farta sheep, while Gulilat et al. [7] obtained an average carcass yield of 10 kg for the same breed. The causes for low performances of indigenous sheep are known to be multilaterally, but largely related to the lack of effective breeding programs. For decades, crossbreeding between exotic sheep (such as Awassi from Israel and Dorper from South Africa) and some indigenous sheep breeds have been performed [8, 9]. However, achievements are far below expectations due to the lack of effective crossbreeding strategies and poor adaptability of crossbreeds that have high exotic blood level [8]. Recently, community-based sheep breeding programs have been designed and implemented for Menz, Afar, Bonga and Horro indigenous sheep breeds [10, 11]. Nevertheless, the majority of the indigenous sheep breeds in different parts of the country are still managed in the traditional breeding system, without being supported by proven scientific methodologies and the state of art technologies in animal breeding.

Given the suitability of the area and adaptive potentials of the existing sheep, improving sheep productivity can be a pathway to put smallholders out of poverty in the northwest highlands of Ethiopia. However, to design and implement effective breeding programs, breeding objective traits specific to sheep breeds reared in the target areas have not yet been identified. In other words, a thorough analysis of sheep breeding experiences and trait preferences is required to sensibly define breeding objectives and design genetic improvement programs at smallholder level [4, 5].

Various methodological approaches have been used to identify breeding objective traits in sheep in Ethiopia, for instance choice experiments [12], ranking among a list of traits [13, 14], and live animal rankings [10, 15]. Recently, a choice experiment has been widely applied to investigate farmers' preferences for animal traits. It provides a hypothetical depiction of attribute levels, giving adequate options for the respondents to reflect their interests. Although it requires higher cognitive efforts, a choice experiment is useful to identify preferences when the number of sheep per household is very small to conduct a live animal ranking method [16]. Overall, applying a combination of methods has been suggested to effectively identify breeding objective traits [17]. As such, this study aimed at identifying breeding practices and smallholder farmers' trait preferences for indigenous sheep in the northwest highlands of Ethiopia, using a ranking of traits and choice experiment methods.

## Materials and methods

### Ethics statement

Prior to the study, data collection formats and procedures were reviewed and approved by the research ethics review committee of Debre Tabor University, Ethiopia (number DTU13/19).

Respondents also provided their verbal informed consent to take part in this study. Furthermore, the data were analyzed anonymously and names, ethnicity and religious issues were not asked and recorded during data collection.

## Description of the study areas

The study was conducted in Estie, Farta and Lay Gayient districts of South Gondar Zone of the Amhara Region, located in the northwest highlands of Ethiopia. These districts were selected because they have huge potentials for sheep production, compared to other eight districts of the zone, for instance, in terms of the availability of communal grazing land and large sheep population. Furthermore, a preliminary assessment prior to the present study has identified the three districts as niche areas for Farta sheep breed, in which designing and implementing a breeding program is under consideration. The agricultural practice in these areas is crop-livestock mixed farming, where livestock play invaluable roles for crop cultivation and the livelihood of smallholder farmers [3].

The detailed comparisons in terms of area coverage, human and livestock populations of the three districts with that of the country, Amhara Region and South Gondar Zone are displayed in Table 1. Concerning climate variables, the average annual rainfall for Estie, Farta and Lay Gayient districts are 1591, 1122 and 1200 mm, respectively. Estie and Lay Gayient districts have received similar minimum (9˚C) and maximum (22˚C) average daily temperatures, whereas the minimum and maximum average daily temperatures of Farta districts are 8˚C and 18˚C, respectively.

## Sample areas and focus-group discussion

Secondary data about livestock population and distribution and availability of infrastructures were obtained from the Agriculture Office of the respected districts. Based on sheep population size and accessibility for transport services, three *kebeles* (the lowest formal administrative units in the district) were selected from each district. For focus-group discussion, a list of smallholder farmers who are regarded as knowledgeable in sheep breeding was identified with the help of livestock development agents and administrative staffs of the *kebele*. Then, three participants per *kebele* were randomly selected from the list. For each district, a separate focus-group discussion was held by involving nine well-experienced farmers, a livestock expert and researcher, with the latter two played facilitation roles. During the discussion, a list of ram and ewe traits was identified taking into account their socio-cultural and economic importance,

**Table 1.  Description of the study districts in comparison to the national, regional and zonal level.**

| | Ethiopia [c] | Amhara Region [c] | South Gondar Zone [c] | Study districts [d] | | |
| --- | --- | --- | --- | --- | --- | --- |
| | | | | Farta | Lay Gayient | Estie |
| **Area** [a] | 426,400 | 59,733.46 | 5442.18 | 424.42 | 587.81 | 527.1 |
| **Human** [b] | 94,351,001 | 21,134,988 | 2,484,929 | 272,177 | 251,926 | 251,708 |
| **Cattle** | 60,392,019 | 16,148,390 | 1,808,185 | 213,188 | 120,579 | 190,853 |
| **Sheep** | 31,302,357 | 11,086,083 | 1,085,652 | 113,978 | 88,836 | 191,985 |
| **Goat** | 32,738,385 | 7,766,661 | 514,746 | 51,556 | 48,758 | 104,604 |

[a] area coverage is in square miles (Source: [18])

[b] human population projection of Ethiopia for 2017 (Source: [19])

[c] cattle, sheep and goat populations of Ethiopia, Amhara Region and South Gondar (Source: [1])

[d] cattle, sheep and goat populations of the three districts (Source: South Gondar Zone Livestock Department annual report for 2017, Unpublished)

both at the local and national levels. To prioritize traits, each participant farmer was asked to rank the traits based on their indigenous knowledge. Finally, the ranks from the three districts were combined and analyzed to identify the most important traits of rams and ewes in which choice experiments were designed.

## Study approach

The study followed two approaches to acquire the data. First, data about socioeconomic status and sheep breeding experience of the respondents were collected through in-person interviews, using a semi-structured questionnaire. The questionnaire was prepared following Haile et al. [16], who suggested the types of information that need to be collected regarding sheep breeding practices and breeding objectives for the purpose of designing a community-based breeding program. The questionnaire was translated into the local language (Amharic). It includes sheep breeding practices such as production objectives, flock size and composition, selection and culling criteria, ram use and mating system. The list of smallholder farmers who own sheep and dwell permanently in the sampled *kebeles* was obtained from the local authorities of the respected *kebeles* of each district. From the list, respondents were selected using a simple random sampling method. For the respondents, willingness and having sheep were the criteria required to involve in the study.

Second, sheep trait preferences were collected by means of a discrete choice experiment, where the respondents have chosen their preferred alternative from choice sets built through hypothetical trait levels combinations. Valuing of non-marketable traits is the typical usefulness of the choice experiment method in animal breeding. Furthermore, in low input production systems, where smallholders' literacy level is low and performance recording is virtually absent, trait preference could be better elucidated using choice experiment method [10, 20]. However, when a choice is made among too many tasks, a choice experiment may not reveal true preferences due to biases allied with choice complexity [21]. Thus, the number of attributes and levels should be manageable in size to minimize the complexity of the choice experiment design.

## Choice experiment design

Identifying attributes and their levels are the principal steps to design choice experiments [22]. In this study, sheep traits were identified and prioritized with focus-group discussions as described earlier. Such an approach has been implemented by Amadou et al. [23] and Siddo et al. [24] in sheep and cattle, respectively. Based on the results of the focus-group discussion, six traits for rams and seven traits for ewes were selected. Attributes with their descriptions and levels are shown in Table 2. Levels were effect-coded to minimize confounding between parameter estimates [22]. A JMP software version 14 [25] was used to construct a full factorial design containing 64 and 128 profiles for rams and ewes, respectively. However, making a choice among the full factorial could be cognitively complex for the smallholder farmers. To reduce the size of the design, a fractional factorial with resolution IV design was applied [26]. Subsequently, an orthogonal array with 16 profiles grouped into four blocks was generated. Furthermore, to prevent forced choices, an opt-out option was assigned in each block, making the number of choice sets five per block.

Choice cards with picture representations of the different profiles of rams and ewes were prepared. Enumerators with a Bachelor's Degree in Animal Science were selected and trained to collect data under the close supervision of the researchers. Prior to the collection of the actual data, the questionnaire and choice cards were pretested in the study sites. Based on the

**Table 2. Ram and ewe attributes and levels included in the choice experiment.**

| Attributes | Attribute descriptions | Levels with effect-coded |
|---|---|---|
| **Body size** | the physical appearances including the height and body length of rams and ewes | Large = 1, small = -1 |
| **Coat color** | the type of color predominantly observed on the body of rams and ewes | brown = 1, white = -1 |
| **Growth** | yearling live weight in which rams and ewes reach at breeding age | Rapid = yearling weight 30 kg = 1, slow = yearling weight 20 kg = -1 |
| **Lambing intervals** * | the average lambing interval between two successive lambing of ewes | short = 3 lambing in 2 years = 1, long = 1 lambing per year = -1 |
| **Mothering ability** * | the ability of ewes to nourish their lambs that could also be implicated on lamb growth and survival | Good = 1, poor = -1 |
| **Litter size** * | the number of lambs born per ewe per lambing | twin = 1, single = -1 |
| **Ear size** $ | the size of the ears of rams | Large = 1, small = -1 |
| **Horn**$ | the presence and absence of horn in rams | Horned = 1, polled = -1 |
| **Tail type** | the length and width of the tail in rams and ewes | length covering the testicular area with sufficient width for rams and extended halfway to the udder for ewes = good = 1, small and thin = bad = -1 |

*appeared only in ewes,

$ appeared only in rams, unmarked attributes appeared in both ewes and rams

feedback, modifications were made on the questionnaire and illustrations of the profiles. A sample of final choice cards used for ewe attributes is shown in Fig 1.

## Sample size

To determine the minimum number of respondents required for the choice experiment, we used the method suggested by Johnson and Orme [27] and Orme [28] as:

$$N > (500 * c)/(t * a) \qquad (1)$$

Where $N$ is the sample size, $c$ is the highest number of levels of any attribute for main effect design, $t$ represents the number of choice tasks, $a$ is the size of choice set per task.

In this study, the number of choice tasks per block was one with four alternatives, excluding the opt-out option and a maximum of two levels per attribute. Plugging in the information to the formula resulted in a minimum sample size of 250. However, we collected the data from a total of 385 respondents randomly selected among smallholder farmers who own sheep.

## Data collection

Procedurally, the respondents were asked first about their socioeconomic characteristics and experiences about sheep breeding in a face-to-face interview. Subsequently, choice cards were introduced. Following a brief explanation about the choice cards, respondents were asked to choose the most preferred hypothetical sheep for breeding purposes. The field data were collected from March to June 2019. From the total respondents, 370 farmers (Estie = 117, Farta = 125 and Lay Gayient = 128) have successfully completed both the questionnaire and the choice experiments. Only 0.47% and 0.4% of the respondents have chosen the opt-out options for rams and ewes, respectively. Such a small proportion of the opt-out options may indicate that the available alternatives are plausible. Given the very small choices, the opt-out options were ignored during data analysis.

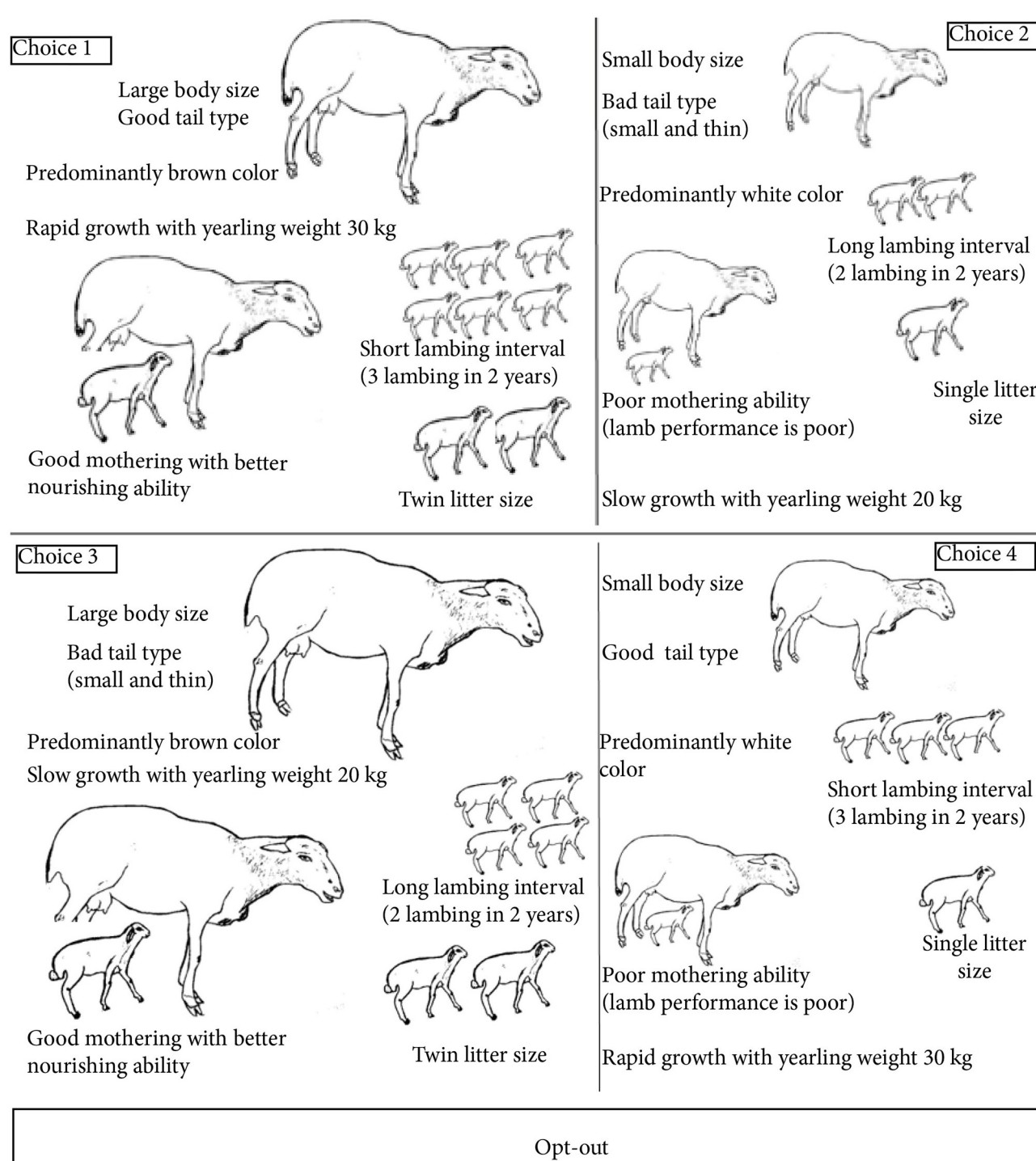

**Fig 1. Sample choice card for ewe attribute levels.**

## Choice modeling and statistical analysis

We started the empirical analysis of preference data using a standard multinomial logit model. Estimation of utility coefficients using a multinomial logit model requires the assumption of independence of irrelevant alternatives (IIA). However, the data could not satisfy the IIA assumption when tested using the Hausman-McFadden test. Fortunately, the generalized multinomial logit (G-MNL) model, derived from mixed and scaled multinomial logit models, does not require the assumption of IIA [29, 30]. Furthermore, the G-MNL model accounts for both taste and scale heterogeneity implying that utility estimates will not be affected by variations in the unobserved component of the model [30]. Mathematically, attribute utility using G-MNL model can be computed as:

$$U_{int} = [\sigma_n \beta + \gamma \eta_n + (1 - \gamma)\sigma_n \eta_n]X_{int} + \varepsilon_{int} \tag{2}$$

Where $U_{int}$ is the utility associated with the $i^{th}$ alternative chosen by the $n^{th}$ respondent ($n = 1,$ ..., 370) at $t$ choice scenario (t = 1 to 4)), $X_{int}$ is a vector of observed attribute levels of ram and ewe, $\beta$ is a vector of mean attribute utility weight, $\eta$ is s a random term associated with a person-specific deviation from the mean utility, $\varepsilon_{int}$ is the idiosyncratic error term, $\sigma_n$ represents the scale of the error term, $\gamma$ is a parameter (value between 0 and 1).

Following Eq 2, let $Ynit = 1$ if the respondent n has chosen the $i^{th}$ alternative at t choice situation, or 0 otherwise, the simulated probability choice in the G-MNL model takes the following form.

$$\hat{P}_n = \frac{1}{D}\sum_{d=1}^{D}\prod_t\prod_i\left(\frac{\exp(\sigma^d\beta + \gamma\eta^d + (1-\gamma)\sigma^d\eta^d)X_{nit}}{\sum_{k=1}^{i}\exp(\sigma^d\beta + \gamma\eta^d + (1-\gamma)\sigma^d\eta^d)X_{nkt}}\right)^{Y_{nit}} \tag{3}$$

Where $\sigma^d = \exp(\overline{\sigma} + \tau\varepsilon_0^d)$, $\eta^d$ is a K-vector distributed with multivariate normal (0, $\Sigma$), $\tau$ is the standard deviation of the scale of the error term, whereas $\varepsilon_0^d$ is a scalar with normal distribution $N(0, 1)$. The simulation process requires d = 1... D draws for $\{\eta^d\}$ and $\{\varepsilon_0^d\}$.

Fiebig et al. [30] proposed different ways of implementing the G-MNL model by imposing restrictions on parameters. In the present study, we set the value $\gamma$ to zero ($\gamma = 0$) to compute the utility coefficient. Given the difficulty of setting an appropriate initial value, most researchers restrict the value of $\gamma$ to zero.

Maximum simulated likelihood estimates, the goodness of fit of the model and the odds ratio were estimated using the function *mlogit* and *gmnl* packages of R software version 3.6.1 [31]. Data for ram and ewe trait preferences were analyzed separately. Socioeconomic characteristics of the respondents, such as sex, educational status, location, sheep flock size, and crop and grazing land sizes, were evaluated for possible influences on preferences of smallholder farmers. However, none of them has exerted a significant effect on trait preferences, thus were omitted from G-MNL to keep models parsimony. Indices for ranking of sheep production objectives and ram and ewe selection criteria of the smallholder farmers were calculated for the first four ranks following König et al. [32] and Bett et al. [33].

$$Index = \sum_{n=1}^{4}a_m X_{nm}\Big/\Big(\sum_m\sum_{n=1}^{4}a_m X_{nm}\Big) \tag{4}$$

Where $a_m$ is the rank weight associated with trait or criteria m ($a_1 = 4$, $a_2 = 3$, $a_3 = 2$, $a_4 = 1$), $X_{nm}$ is the proportion of smallholder farmers who ranked the $m^{th}$ trait or criteria in the $n^{th}$ rank (n = 1 to 4 ranks), $m$ represents sheep traits and the different purposes of sheep production.

## Results

### General characteristics of the respondents

From a total of 370 respondents, about 95.4% were male farmers. Such a large variation in gender participation was observed because the number of female-headed households who owned sheep and available for sampling during the study period was very small. All the respondents stated that their main occupation is agriculture, involving both crop and livestock productions. The mean crop and private grazing landholdings were 0.8 and 0.28 hectares, respectively. With respect to the literacy status of the respondents, 32.7% had attended informal and religious schools and were able to read and write. Likewise, about 29.2% of the respondents had attended primary school, while only 11.6% reached a post-primary school level. The remaining 26.5% of the respondents were illiterate.

### Sheep flock size and composition

Mean flock size and standard deviations for each category of sheep and sampling districts are given in Table 3. The overall mean for sheep flock size of the study areas was 10.21, with a flock size range of 2 to 43 sheep. Across the study districts, sheep flock size was significantly larger in Lay Gayient than the two other districts (P<0.001). With the overall mean of 5.21 ewes, the proportion of breeding ewes accounted for about 51% of the total flock size. Moreover, a significant difference exists among the three districts in the mean number of breeding ewes (P<0.001). Given the overall mean of 0.48, the number of breeding rams was generally small across the study districts. The ratio of breeding ram to ewes was 1:11. Within the flock, ewe and ram lambs are known to be used as replacements for breeding flock or sold for income generation or otherwise slaughtered for consumption. Lambs account for about 26.6% of the total sheep flock.

### Sheep production objectives

The importance of sheep production for the smallholder farmers was examined based on the overall index calculated from the proportions of four ranks (Table 4). Given the highest index value of 0.36, income generation was ranked as the primary reason for keeping sheep. In

**Table 3. Sheep flock size and composition at the smallholder level by the study districts.**

| Sheep category$ | Districts+ | | | | | | Overall (N = 370)* | |
|---|---|---|---|---|---|---|---|---|
| | Estie (N = 117) | | Farta (N = 125) | | Lay Gayient (N = 128) | | | |
| | Mean | Std. | Mean | Std.# | Mean | Std. | Mean | Std. |
| **Breeding ewes** | 5.15[a] | 1.34 | 3.86[b] | 1.19 | 6.64[c] | 3.2 | 5.23 | 2.43 |
| **Breeding rams** | 0.52[a] | 0.57 | 0.46[a] | 0.5 | 0.45[a] | 0.6 | 0.48 | 0.56 |
| **Ewe lambs** | 0.45[a] | 1.03 | 0.64[a] | 0.87 | 1.87[b] | 1.53 | 1.01 | 1.34 |
| **Ram lambs** | 0.62[a] | 0.94 | 0.79[a] | 0.92 | 0.68[a] | 1.15 | 0.7 | 1.01 |
| **Lambs** | 2.83[a] | 1.33 | 2.76[a] | 1.75 | 2.58[a] | 2.16 | 2.72 | 1.79 |
| **Castrated** | 0.07[a] | 0.31 | 0.03[a] | 0.18 | 0.14[a] | 0.68 | 0.08 | 0.45 |
| **Total flock size** | 9.65[a] | 2.76 | 8.54[a] | 3.4 | 12.36[b] | 6.39 | 10.21 | 4.79 |

*N is the number of respondents,

#Std. is the standard deviation,

+Means in a row with different letters are significantly different (P<0.001)

$Ewe and ram lambs represent those with age group between six months and one year while lambs are both male and female groups with age below six months

**Table 4. Rank proportions and index values for sheep production objectives at smallholder level.**

| Sheep production objectives | Rank proportions* | | | | Index |
|---|---|---|---|---|---|
| | **1** | **2** | **3** | **4** | |
| Income generation | 0.65 | 0.33 | 0.01 | 0.01 | 0.36 |
| Meat source | 0.32 | 0.52 | 0.14 | 0.03 | 0.31 |
| Saving/asset | 0.02 | 0.08 | 0.18 | 0.09 | 0.08 |
| Manure source | 0.00 | 0.07 | 0.45 | 0.44 | 0.16 |
| Sheep skin source | 0.00 | 0.01 | 0.22 | 0.42 | 0.09 |
| Wool production | 0.00 | 0.00 | 0.01 | 0.03 | 0.00 |

*1–4 represent rank 1, rank 2, rank 3 and rank 4

addition, meat for consumption purpose and manure as organic fertilizer and local fuel sources were ranked as second and third objectives, respectively. Wool production was the least ranked objective of sheep production. However, nearly all respondents practiced the shearing of sheep at least once per year as part of controlling external parasites and have used the wool for traditional purposes.

## Ram use practices and mating management

Across the study districts, about 46.2% of the smallholder farmers have their own breeding rams with sources born in the flock (36.8%), purchased with partners (7.8%) and bought privately (1.6%). On the other hand, more than half (53.8%) of the respondents were without breeding rams, but all had access to ram services from their neighbors and relatives. According to the view of the respondents, the average duration in which a particular ram stayed within the flock for breeding purposes is about 2.26 years. All the respondents stated that mating is uncontrolled with year-round lambing. Given the absence of pedigree records coupled with the practices of using rams born within the flock, the level of inbreeding within the flock is expected to be high. While discussing with smallholder farmers, their understanding of inbreeding appeared to be minimal.

## Ram and ewe selection criteria

About 91% of the respondents have been practicing selection of breeding rams and ewes within their own flock using different criteria. Index values calculated for smallholder farmers' selection criteria for breeding rams and ewes are given in Table 5. Given the highest index value of 0.34, body size was ranked as the primary selection criteria for breeding rams, followed by coat color and tail type. Similarly, body size (index = 0.34) and coat color (index = 0.25) were two of the most important breeding ewe selection criteria. However, records about sheep performance and pedigree were nonexistent in all respondents involved in the study, thus selection decisions are mainly based on physically observable attributes of sheep. Given the very low index values, adaptation and wool yield traits were the least considered selection criteria for both sexes. Across the study districts, there was no disparity in the order of major selection criteria for both rams and ewes.

## Culling criteria for rams and ewes

Nearly all smallholder farmers (97%) responded that both rams and ewes with small body sizes are not preferred for breeding purposes, thus excluded from the flock. Likewise, sheep with

**Table 5. Rank proportions and index values for breeding ram and ewe selection criteria.**

| Selection criteria | Rank proportions and index (rams)* | | | | | Rank proportions and index (ewes) | | | | |
|---|---|---|---|---|---|---|---|---|---|---|
| | 1 | 2 | 3 | 4 | Index | 1 | 2 | 3 | 4 | Index |
| Body size | 0.64 | 0.21 | 0.09 | 0.02 | 0.34 | 0.63 | 0.26 | 0.05 | 0.02 | 0.34 |
| Litter size | - | - | - | - | - | 0.02 | 0.02 | 0.09 | 0.12 | 0.04 |
| Lamb survival | - | - | - | - | - | 0.00 | 0.00 | 0.01 | 0.05 | 0.01 |
| Lamb growth | - | - | - | - | - | 0.02 | 0.04 | 0.03 | 0.08 | 0.04 |
| Lambing interval | - | - | - | - | - | 0.02 | 0.05 | 0.13 | 0.21 | 0.07 |
| Ear size | 0.03 | 0.05 | 0.13 | 0.19 | 0.07 | 0.02 | 0.04 | 0.10 | 0.07 | 0.05 |
| Pedigree | 0.03 | 0.01 | 0.05 | 0.08 | 0.03 | 0.05 | 0.01 | 0.03 | 0.05 | 0.03 |
| Coat color | 0.25 | 0.44 | 0.16 | 0.08 | 0.27 | 0.22 | 0.41 | 0.15 | 0.06 | 0.25 |
| Growth rate | 0.01 | 0.04 | 0.07 | 0.08 | 0.04 | 0.01 | 0.04 | 0.08 | 0.05 | 0.04 |
| Sexual maturity | 0.00 | 0.02 | 0.02 | 0.10 | 0.02 | 0.00 | 0.06 | 0.19 | 0.12 | 0.07 |
| Libido | 0.01 | 0.03 | 0.10 | 0.18 | 0.05 | - | - | - | - | - |
| Tail type | 0.03 | 0.14 | 0.18 | 0.17 | 0.11 | 0.01 | 0.06 | 0.08 | 0.11 | 0.05 |
| Adaptation | 0.00 | 0.00 | 0.01 | 0.01 | 0.00 | 0.00 | 0.00 | 0.02 | 0.02 | 0.01 |
| Wool yield | 0.00 | 0.01 | 0.02 | 0.01 | 0.01 | 0.00 | 0.01 | 0.02 | 0.02 | 0.01 |
| Horn status | 0.00 | 0.02 | 0.02 | 0.10 | 0.06 | 0.00 | 0.00 | 0.01 | 0.00 | 0.00 |

*1–4 represent rank 1, rank 2, rank 3 and rank 4

black coat color were not favored for breeding purposes in 94% of the smallholder farmers who practiced culling due to unwanted coat color. Furthermore, the majority of the respondents said that rams and ewes could be culled due to old age and fertility problems. Based on the view of the respondent, the average culling age for breeding ewes due to oldness was 9.78 years, while rams were excluded from the flock much earlier than becoming old. Moreover, poor mothering ability of ewes implicated largely on lambs' performance was reported to be one of the main culling reasons for female sheep. Conversely, the majority of the respondents did not cull rams and ewes due to poor body condition. The respondents highlighted that body condition fluctuation due to seasonal variations in the availability of feed is very common in the study areas.

## Choice preference for ram and ewe attributes

Utility coefficients estimated for ram and ewe attributes are presented in Table 6. Pseudo R-square ($\rho^2$) estimates of the model were 0.2 and 0.27 for ram and ewes, respectively, implying that the overall fit of the model is good. According to McFadden [34], $\rho^2$ values ranging from 0.2 to 0.4 are indicators for the excellent fit of a model. For each attribute, one of the levels was used as a reference with a value set to zero. Furthermore, body size and tail type for rams, and body size, growth rate and lambing intervals for ewes were fitted as random parameters. The selection of random parameters was based on the overall contribution to model fitness [29].

All attributes of breeding rams and ewes, included in the model as fixed and random parameters (Table 6), have shown significant influences on smallholder farmers' preference (P<0.05). In other words, smallholder farmers preferred a ram with good tail type, large body size, predominantly brown coat color and rapid growth in the presence of horns. However, large ear size with an estimated coefficient of -0.1869 was not preferred over a ram with small ears. The magnitude of the utility coefficient shows that tail type of the ram is the most preferred attribute, while ear size is the least preferred. Regarding the choices for breeding ewes, all attributes resulted in positive utility coefficients as expected. Based on the magnitude of

**Table 6. Estimates of smallholder farmers' preferences for breeding ram and ewe attributes.**

| | G-MNL model (ram) | | | G-MNL model (ewe) | | |
|---|---|---|---|---|---|---|
| | Estimates | SE | P-value | Estimates | SE | P-value |
| **Fixed parameters** | | | | | | |
| **1: Intercept*** | 0.0 | - | - | 0.0 | - | - |
| **2: Intercept** | 0.2863 | 0.117 | 0.014 | 1.1342 | .251 | <0.001 |
| **3: Intercept** | -0.9403 | 0.163 | <0.001 | 0.6060 | 0.236 | 0.01 |
| **4: Intercept** | 0.9929 | 0.152 | <0.001 | 1.0351 | 0.241 | <0.001 |
| **Coat color (brown)** | 0.3237 | 0.039 | <0.001 | 1.1517 | 0.312 | <0.001 |
| **Growth rate (rapid)** | 0.2666 | 0.043 | <0.001 | - | - | - |
| **Ear size (large)** | -0.1869 | 0.060 | 0.002 | - | - | - |
| **Horn (horned)** | 0.2523 | 0.041 | <0.001 | - | - | - |
| **Mothering ability (good)** | - | - | - | 1.3056 | 0.357 | <0.001 |
| **Litter size (twin)** | - | - | - | 0.7737 | 0.327 | 0.018 |
| **Tail type (good)** | - | - | - | 0.8547 | 0.240 | <0.001 |
| **Random parameters** | | | | | | |
| **Body size (large)** | 0.5132 | 0.048 | <0.001 | 1.1287 | 0.333 | <0.001 |
| **Tail type (good)** | 0.8986 | 0.051 | <0.001 | - | - | - |
| **Growth rate (rapid)** | - | - | - | 0.9618 | 0.254 | <0.001 |
| **lambing interval (short)** | - | - | - | 0.9645 | 0.23 | <0.001 |
| **The standard deviation of random parameters** | | | | | | |
| **Body size (large)** | 0.001 | 0.072 | 0.99 | 0.0055 | 0.07 | 0.94 |
| **Tail type (good)** | 0.014 | 0.318 | 0.96 | - | - | - |
| **Growth rate (rapid)** | - | - | - | 0.2086 | 0.244 | 0.39 |
| **lambing interval (short)** | - | - | - | 0.0745 | 0.171 | 0.66 |
| **Tau (τ)** | 0.004 | 0.13 | 0.96 | 0.7565 | 0.172 | <0.001 |
| **Gamma (γ)** | 0 | - | fixed | 0 | - | fixed |
| **Log likelihood null** | -1864.1 | - | - | -1894.4 | - | - |
| **Log likelihood function** | -1493.1 | - | - | -1376.3 | - | - |
| **Number of observations** | 1473 | | | 1474 | - | - |
| **McFadden's $\rho^2$** | 0.20 | - | - | 0.27 | - | - |

*the first choice alternative was used as a reference category hence intercept is zero

utility coefficients, good mothering ability was the most preferred ewe attributes, followed by predominantly brown coat color and large body size. Likewise, short lambing interval and rapid growth rate were the fourth and fifth preferred attributes, while good tail type and twinning were placed in the bottom as the least preferred ewe attributes.

Estimated coefficients for the standard deviation of random parameters for both rams and ewes were not significant (P>0.05), indicating that significant preference heterogeneity was not observed among respondents (Table 6). In contrast to the choice situations for ram attributes, the standard deviation of the scale parameter (τ = 0.7565) revealed significant scale heterogeneity for ewe attribute choice scenarios (P<0.001).

## The odds ratio for choosing ram and ewe attribute levels

The odds of selecting a ram with good tail type is 2.46 times higher than a ram with bad tail type (Table 7). Similarly, a ram with large body size is 1.67 times more likely to be chosen by smallholder farmers than a ram with small body size. However, the choice of a ram having large ear size is less likely than a ram with small ear size. Concerning the odds of ewe attribute

**Table 7. The odds ratios for all levels of the different ram and ewe attributes.**

| Attributes | G-MNL model | |
|---|---|---|
| | **Ram odd ratio (95% CI)*** | **Ewe odd ratio (95% CI)** |
| **Body size (large vs small)** | 1.67 (1.52 to 1.84) | 3.09 (1.61 to 5.94) |
| **Coat color (brown vs white)** | 1.38 (1.28 to 1.49) | 3.16 (1.72 to 5.83) |
| **Growth rate (rapid vs slow)** | 1.31 (1.20 to 1.42) | 2.62 (1.59 to 4.31) |
| **Tail type (good vs bad)** | 2.46 (2.22 to 2.71) | 2.35 (1.47 to 3.77) |
| **Ear size (large vs small)** | 0.83 (.74 to .93) | - |
| **Horn status (horned vs polled)** | 1.29 (1.19 to 1.40) | - |
| **Lambing interval (short vs long)** | - | 2.62 (1.67 to 4.12) |
| **Mothering ability (good vs poor)** | - | 3.69(1.83 to 7.42) |
| **Twining (twin vs single)** | - | 2.17 (1.14 to 4.12) |

*95% lower and upper confidence intervals in the parenthesis

levels, ewes with good mothering ability is 3.69 times more likely to be picked up by small-holder farmers than ewes with poor mothering ability. Overall, all the preferred attribute levels of ewes have shown high odds ratios compared with the reference level of the corresponding attributes.

## Discussion

A good understanding of livestock production and the breeding system is fundamental to design a breeding program at the smallholder level [35, 36]. More importantly, for breeding programs to be operational at smallholder levels, the active involvement of smallholder farmers from planning to implementation is strongly advised by multiple scholars [13, 32, 37]. In this study, sheep breeding practices and trait preferences of smallholder farmers were identified using participatory approaches. The results of the present study can be used to design village-based breeding programs for indigenous sheep inhabiting the northwest highlands of Ethiopia.

### Sheep production objectives and flock structure

In the study areas, income generation and meat consumption were the main objectives of keeping sheep by smallholder farmers. Such objectives imply that indigenous sheep can play huge roles in poverty reduction at the smallholder level if adequate efforts are made to improve their productivity. Similar sheep production objectives have been reported in different parts of Ethiopia, particularly in the crop-livestock mixed farming system [4, 5, 14, 38].

The present study revealed that the average flock size per smallholder farmer is generally small. In-country wise, sheep flock size per household is reported to vary across production systems depending on the availability of inputs and dependence on livestock. For instance, Edea et al. [14] reported sheep flock size of 8 to 11 in a crop-livestock mixed production system, while Nigussie et al. [39] reported an average flock size of about 97 and 72 sheep in Pastoral and agro-pastoral production systems, respectively. For breeding programs targeting genetic improvement through selection, the small number of sheep available at the individual farmer level could be problematic, for instance, from the perspectives of minimizing inbreeding and obtaining optimal genetic gain. In such scenarios, establishing strong collaboration among smallholder farmers to create large breeding flock is fundamental for the sustainable use of genetic resources.

## Sheep breeding practices

Valuing indigenous knowledge is vital to ensure the sustainability of a breeding program intended to be implemented at the smallholder level [36, 37]. Interestingly, the majority of the smallholder farmers in the study areas have been practicing selection within their own sheep flock using diverse criteria. It was found that body size and coat color were two of the most important selection criteria for both breeding rams and ewes. Biologically, coat color has a qualitative nature, implying that it cannot be measured on a scale basis. Large body size, on the other hand, can be expressed in terms of linear body measurements such as body length, height, pelvic and girth circumferences. In addition, linearly measured body size traits are easy to measure even at smallholder level and are reported to be moderately heritable [40]. Such facts imply that smallholder farmers could indeed achieve some level of improvement when applying selection based on body size.

Furthermore, although the growth rate was not ranked among the top selection criteria in both sexes, it has been known to have moderate to high genetic and phenotypic correlations with body size traits that can be measured linearly [40, 41]. Thus, body size based selection practices of the smallholder farmers could have a positive impact on the growth rate. With regard to the other attributes, sheep breeds in the study areas are naturally characterized by a fatty tail type, horizontally orientated ear and males are often horned [2, 42].

Another important experience of smallholder farmers in the study areas is the practice of culling of sheep perceived as not suitable for breeding purposes within their own flock. Reasons for culling were due to small body size, unfavorable coat color, old age and fertility problems for both male and female sheep, and due to poor mothering ability of ewes. Such practices are also reported to be implemented by smallholder farmers in other parts of Ethiopia [43].

In the study areas, smallholder farmers often keep breeding ewes for longer periods, while breeding rams stay within the flock for a relatively short period. The practice of using rams for a short duration is assumed to be useful to minimize inbreeding within the flock. Yet, a large number of the smallholder farmers used rams born in the flock, that could result in mating between genetically related sheep, thereby increasing inbreeding. The issue of inbreeding at smallholder levels has been also indicated in other parts of Ethiopia [4, 14, 43] and elsewhere in West Africa [44]. Although pedigree-based records are lacking, the practice of random mating and the use of communal rams can be considered as encouraging breeding practices to reduce the effect of inbreeding in the study areas.

## Choice preference for sheep traits

Choice modeling was applied to elucidate the preference of smallholder farmers for indigenous sheep traits. Among ram attributes, a good tail type was the most preferred attribute, followed by large body size and predominantly brown coat color. These attributes were also among the major ram selection criteria, although the order of importance is reversed. The choice of ram attributes seemed to be a reflection of the existing traditional breeding approach implemented in the study areas. Duguma et al. [12] performed a choice modeling in four Ethiopian indigenous sheep breeds (Horro, Menz, Afar and Bonga). These authors reported that sexual activity (libido) for Horro and Menz, good tail type for Bonga and color type for Afar sheep breeds were the most preferred ram attributes. Although libido attribute was not included in the choice experiment of the present study, it was ranked as the fifth ram selection criteria by the smallholder farmers (Table 5). Elsewhere in West Africa, Tindano et al. [45] have reported a high preference for disease resistance in rams. Such preference heterogeneity for ram attributes could likely be due to differences in sheep breed, production system and sociocultural characteristics of smallholder farmers.

Rapid growth rate and the presence of horns were also found to be important ram attributes given their significant influence on the preference of smallholder farmers. Growth related traits, such as weight at different ages, are easy to measure and have reasonably high heritability. Due to such characteristics, growth traits have been the main targets of breeding programs piloted for a few indigenous sheep breeds in Ethiopia [46, 47]. Furthermore, the earlier attempts of selection for body weight on nucleus herds have shown that Ethiopian indigenous sheep are reasonably responsive for selective breeding [48]. Unexpectedly, a ram with small ear size was preferred to a ram with larger ears, although ear size is the least important attribute in terms of the magnitude of the utility coefficient. Based on the index value in Table 5, ear size was ranked as the fourth ram selection criteria, implying that the emphasis of smallholder farmers toward ear size is not strong.

Regarding the preference for ewe attributes, good mothering ability, representing the nourishing potential of ewes for better growth and survival of lambs, was the most preferred attribute. This result is in agreement with Duguma et al. [12] who reported a high choice preference for good mothering ability of ewes in four indigenous sheep breeds of Ethiopia. The strong preference for mothering quality could be a good indicator of the mechanism by which smallholder farmers are trying to be profitable, even under low input sheep production systems. One reason could be is that smallholder farmers often sell lambs for income generation, thus well-nourished lambs are expected to fetch a better price.

Although the tail type and litter size of ewes had shown significant influences on the preference of smallholder farmers, both were less important compared to other ewe attributes. This could indicate that the main emphasis of smallholder farmers for ewe choice is lamb production while focused largely on observable characteristics for ram profiles. However, for ewe attributes other than good mothering ability, smallholder farmers' preference obtained in the present study was not in agreement with the findings of Duguma et al. [12]. This clearly highlights the importance of evaluating sheep trait preferences on a breed basis prior to designing a breeding program.

One appealing feature of choice modeling using G-MNL is the possibility of accommodating choice heterogeneity among respondents. Unlike for the breeding ram, ewe preference analysis revealed significant scale heterogeneity. A scale heterogeneity could occur due to variations in the choice behavior of respondents [30]. For instance, for some respondents, the choice may be driven by one or a few attributes resulting in a very small estimate for the scale of the error term or a very large utility coefficient. Such choice characteristics could likely be the source of scale heterogeneity in the present study. Fortunately, the G-MNL model takes into account both taste and scale heterogeneity simultaneously. Thus, maximum simulated likelihood estimates of parameters will not be biased due to heterogeneity.

## Potential traits for breeding objective

The present study revealed that smallholder farmers have given due emphasis for both quantitative and qualitative attributes of indigenous sheep. Given the easiness of measurement and heritable nature of the traits, body size and growth rate can be the main components of sheep breeding objectives in the study areas. Linear body measurements such as body length, chest girth, pelvic width and height at wither are proxy traits for body size, while growth can effectively be described in terms of weight at different ages. Furthermore, lambing interval, early growth and survival of lambs should be incorporated in the breeding objective. Although the latter two traits were not among the main ewe selection criteria, they are more likely to reflect the mothering ability of ewes that highly influenced smallholder farmers' preference.

However, qualitative traits such as tail type and coat color cannot be measured on a scale basis, thus are difficult to incorporate directly in the breeding objective, despite their significant effects on preference. This implies selection decisions at smallholder level should not only be based on estimated breeding value of traits in the breeding objective but instead, additional preferences of farmers should be taken into account, for example, in addition to estimated breeding values, considering coat color and tail type as selection decision criteria. It has been said that more than any form of financial support, satisfying the interest of smallholder farmers is vital for the sustainability of genetic improvement programs [37]. Overall, production, reproduction and adaptive traits could be combined alternatively for optimal genetic gain without making the breeding objective more complex.

## Conclusion

Although average flock size per head is small, the majority of the smallholder farmers have experiences in selecting and culling of rams and ewes within their own sheep flock. Given the absence of any form of sheep performance recording system at the smallholder level, the breeding practices and decisions mainly rely on observable characteristics of sheep. Based on the magnitude of utility coefficients, a good tail type was the most preferred ram attribute followed by large body size, predominantly brown coat color and rapid growth rate. Similarly, smallholder farmers have shown their highest preference for good mothering ability of ewes followed by predominantly brown coat color, large body size, short lambing interval, and rapid growth rate. The present results implied that breeding objectives incorporating production, reproduction and adaptive sheep traits can be derived by alternatively combining highly preferred attributes having measurable and heritable characteristics. However, selection decisions at the smallholder level should not only be based on the outcome of traits included in the breeding objective but instead, additional preferences of farmers need to be taken into account.

## Supporting information

**S1 File. Raw data for respondents' general characteristics and sheep breeding practices.** It is an SPSS file with all the variables explained fully in the variable view section.
(SAV)

**S2 File. Raw data for breeding ram and ewe choice preferences.** It is an Excel file with all ram and ewe variables explained fully in the third cell of the sheet.
(XLSX)

**S3 File. R script for breeding ram and ewe choice preference data analysis.** R script for ram and ewe choice preferences are available separately but contained in the same file.
(TXT)

**S4 File. The questionnaire used in the survey (English version).** The questionnaire developed to collected socioeconomic and sheep breeding practice data.
(PDF)

**S5 File. The questionnaire used in the survey (Amharic version).** The questionnaire translated from English into the local language (Amharic).
(PDF)

## Acknowledgments

The corresponding author would like to thank smallholder farmers from Estie, Farta and Lay Gayient districts who spent their invaluable time to provide the required data for this study.

## Author Contributions

**Conceptualization:** Abiye Shenkut Abebe, Kefyalew Alemayehu, Anna Maria Johansson, Solomon Gizaw.

**Data curation:** Abiye Shenkut Abebe.

**Formal analysis:** Abiye Shenkut Abebe.

**Funding acquisition:** Abiye Shenkut Abebe.

**Investigation:** Abiye Shenkut Abebe.

**Methodology:** Abiye Shenkut Abebe.

**Supervision:** Kefyalew Alemayehu.

**Validation:** Kefyalew Alemayehu.

**Writing – original draft:** Abiye Shenkut Abebe.

**Writing – review & editing:** Kefyalew Alemayehu, Anna Maria Johansson.

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
