## [Decision Letter · Decision Letter 0]

24 Feb 2020

PONE-D-20-02104

Breeding practices and trait preferences of smallholder farmers for indigenous sheep in the northwest highlands of Ethiopia: Inputs to design a breeding program

PLOS ONE

Dear Dr Abebe

Thank you for submitting your manuscript to PLOS ONE. After careful consideration, we feel that it has merit but does not fully meet PLOS ONE’s publication criteria as it currently stands. Therefore, we invite you to submit a revised version of the manuscript that addresses the points raised during the review process.

Many thanks for submitting your manuscript to PLOS One

It was reviewed by two expert reviewers who have recommended some minor revisions be made prior to its publication

If you could address the reviewers comments and write a response to reviewers, this will expedite second review

I wish you the best of luck with your revisions

Many thanks

Simon

We would appreciate receiving your revised manuscript by Apr 09 2020 11:59PM. To enhance the reproducibility of your results, we recommend that if applicable you deposit your laboratory protocols in protocols.io, where a protocol can be assigned its own identifier (DOI) such that it can be cited independently in the future. For instructions see: http://journals.plos.org/plosone/s/submission-guidelines#loc-laboratory-protocols

We look forward to receiving your revised manuscript.

Kind regards,

Simon Russell Clegg, PhD

Academic Editor

PLOS ONE

Journal Requirements:

3. We note that Figure #1 in your submission contain [map/satellite] images which may be copyrighted. All PLOS content is published under the Creative Commons Attribution License (CC BY 4.0), which means that the manuscript, images, and Supporting Information files will be freely available online, and any third party is permitted to access, download, copy, distribute, and use these materials in any way, even commercially, with proper attribution. For these reasons, we cannot publish previously copyrighted maps or satellite images created using proprietary data, such as Google software (Google Maps, Street View, and Earth). For more information, see our copyright guidelines: http://journals.plos.org/plosone/s/licenses-and-copyright.

1.    You may seek permission from the original copyright holder of Figure #1 to publish the content specifically under the CC BY 4.0 license. 

Reviewers' comments:

Reviewer's Responses to Questions

**Comments to the Author**

1. Is the manuscript technically sound, and do the data support the conclusions?

Reviewer #1: Yes

Reviewer #2: Partly

2. Has the statistical analysis been performed appropriately and rigorously? 

Reviewer #1: Yes

Reviewer #2: Yes

3. Have the authors made all data underlying the findings in their manuscript fully available?

Reviewer #1: Yes

Reviewer #2: Yes

4. Is the manuscript presented in an intelligible fashion and written in standard English?

Reviewer #1: Yes

Reviewer #2: Yes

5. Review Comments to the Author

Reviewer #1: The manuscript aimed identifying breeding practices and smallholder farmers’ trait preferences for indigenous sheep in Ethiopia. This kind of study is important since smallholder practices of animal breeding requires different approaches and information are scarce in the literature. The manuscript are well written, but some points need better understanding (attached I provide some comments in the own text). The discussion needs more attention since repeat several times the results. I suggest review it.

Reviewer #2: The manuscript was well-written. Topic is very unique. I suggest to publish the manusript after minor revision.

Introduction

-I would suggest shortening the introduction in term of general context, and including more information about Breeding practices and breeding program for indigenous sheep in Ethiopia.

- I would suggest describing what kind of research work/studies has been done with various; methodological approaches are available to identify breeding objective traits in sheep in Ethiopia.

Materials and methods

- The survey area: It is good to have the map but also add some statistics. How many square miles, relative size to the whole of Ethiopia, what is the population of people, livestock species in the region of study versus the whole country etc.

- How were the sampled villages/ districts chosen? Give more details. What information was there before hand?

-The quality of Figure 2 is not good. Moreover, for the method choice scenario, it is advisable to use 2 choice scenarios and a third scenario called "no choice". How can you further explain your approach and justify its limitations?

-Explain further, how you conducted your focus group discussions.

-You can use the articles below to enrich the Materials & Methods section and the discussion section:

1. Traoré, B., Govoeyi, B., Hamadou, I. et al. Analysis of preferences of agro-pastoralists for the attributes of traction dromedaries in harness cultivation: A case study of the Koro district of Mali. Pastoralism 9, 19 (2019). https://doi.org/10.1186/s13570-019-0153-9

2. Tindano, K., Moula, N., Traoré, A. et al. Assessing the diversity of preferences of suburban smallholder sheep keepers for breeding rams in Ouagadougou, Burkina Faso. Trop Anim Health Prod 49, 1187–1193 (2017). https://doi.org/10.1007/s11250-017-1315-7

3. Siddo, S, Moula, N, Hamadou, I, Issa, M, Marichatou, H, Leroy, P and Antoine-Moussiaux, N 2015. Breeding criteria and willingness to pay for improved Azawak zebu sires in Niger. Archive of Animal Breeding 58, 251–259.

4. Issa Hamadou, Nassim Moula, Seyni Siddo, Moumouni Issa, Hamani Marichatou, Pascal Leroy, and Nicolas Antoine-Moussiaux. Valuing breeders' preferences in the conservation of the Koundoum sheep in Niger by multi-attribute analysis. https://www.arch-anim-breed.net/62/537/2019/aab-62-537-2019.html

-I suggested to include the survey questionnaire.

Results

- The chaper was very well described.

Discussion:

- The chaper was very well described.

-The conclusion should be developed on the outlook for your work.

-The paper is very interetsing, but it can still be improved by exploiting all the collected data.

6. PLOS authors have the option to publish the peer review history of their article (what does this mean?). If published, this will include your full peer review and any attached files.

Reviewer #1: No

Reviewer #2: No

---

## [Author Response · Author response to Decision Letter 0]

11 Apr 2020

Comments from the academic editor:

 Journal Requirements:

Response: Thank you for reminding the very important issue. The authors have prepared the manuscript based on the guideline of PLOS ONE and tried all the best to meet all the PLOS ONE journal style requirements. In all tables footnotes, we changed the italic fonts to normal. We also corrected the file naming for Supporting information files. Ethics statement was added in line 97-102

Response: Thank for raising this issue. We have developed a questionnaire for our study and attached it as supporting information (both in English and local language (Amharic)). While preparing the questionnaire, we followed the suggestions by Haile et al. (2011) regarding what to collect when planning to study sheep breeding practice and identify breeding objectives that can ultimately be used to design sheep breeding programs at smallholder level in Ethiopia condition. The questionnaire doesn’t contain copyrighted material nor owned by someone. 

Furthermore, we added the following text about the survey from line 145 -154

“The questionnaire was prepared following Haile et al. [16] who suggested the types of information that need to be collected regarding sheep breeding practice and breeding objective for the purpose of designing a community-based breeding program. The questionnaire was translated into the local language (Amharic). It includes sheep breeding practices such as production objectives, flock size and composition, selection and culling criteria, ram use and mating system. The list of smallholder farmers who own sheep and dwell permanently in the sampled kebeles was obtained from the local authorities of the respected kebeles of each district. From the list, respondents were selected using a simple random sampling method. For the respondents, willingness and having sheep were the criteria required to involve in the study.”

3. We note that Figure #1 in your submission contain [map/satellite] images which may be copyrighted. All PLOS content is published under the Creative Commons Attribution License (CC BY 4.0), which means that the manuscript, images, and Supporting Information files will be freely available online, and any third party is permitted to access, download, copy, distribute, and use these materials in any way, even commercially, with proper attribution. For these reasons, we cannot publish previously copyrighted maps or satellite images created using proprietary data, such as Google software (Google Maps, Street View, and Earth). For more information, see our copyright guidelines: http://journals.plos.org/plosone/s/licenses-and-copyright.

You may seek permission from the original copyright holder of Figure #1 to publish the content specifically under the CC BY 4.0 license. 

 Responses: Thank you for your concern. The authors agreed to remove figure #1 from the submission. However, what we would like to make clear is that we prepared the map using satellite image data obtained from the Ethiopian Mapping Agency. We contacted the Ethiopian Mapping Agency for possible written permission but told us that such enquire is not common. Otherwise, we haven’t found someone who published and owned the copyright for the map we submitted. Because of such confusion, we decided to remove the figure from the system. We provided detailed descriptions of the study areas in the text and table form (considering the comments given by one of the reviewers).

Comments from reviewers

Reviewer #1

Comments:

The manuscript aimed at identifying breeding practices and smallholder farmers’ trait preferences for indigenous sheep in Ethiopia. This kind of study is important since smallholder practices of animal breeding requires different approaches and information are scarce in the literature. 

The manuscript are well written, but some points need better understanding (attached I provide some comments in the own text). 

Responses: Thank you for highlighting the points and we addressed the feedback line by line as follow:

Line 26, remove the phrase “Choice modeling using”

Response: Line 26, we removed the phrase “Choice modeling using” and corrected the sentence. 

Line 55-56, “risk” is a generic word, clarify the type of risk and “living banks” not an appropriate term, clarify this?

Response: the sentence from line 54-56 was rewritten as follow:

“Particularly when crop farming is less reliable due to drought or other factors, sheep are commonly used to mitigate adverse effects, for instance, related to food shortage at the smallholder level”

Line 57, what is abundances? You used many generic terms. Should be more specific

Responses: the authors acknowledged the reviewer for raising this issue. The term abundance is replaced with “the presence of a large number of sheep” in line 57 and the sentence is corrected accordingly. 

Line 78 – 80, exclude and rearrange some words from the sentence

Response: the sentence from line 78-80 was corrected based on the suggestion as follow:

“Given the suitability of the area and adaptive potentials of the existing sheep, improving sheep productivity can be a pathway to put smallholders out of poverty in the northwest highlands of Ethiopia.”

Line 179, levels for coat colors, has the breed have brown and white colors only?

Responses: The authors thank the reviewer for raising this issue. Indeed, these are not the only coat colors for the sheep breed in the study areas but are the major representative colors. We defined the levels as “predominantly brown” (which includes brown color only, dominantly brown with white, dominantly brown with black color, etc.) and the same thing for the level “predominantly white”. Besides, the proportion of sheep with dominantly black coat color is very small in the sheep population of the study areas <4% (Bimerow et al. 2011). Moreover, having black colored sheep in the flock is considered by many smallholder farmers as a sign of misfortune implying that incorporating, for instance, black color as the third level could easily lead lexicographic behavior among farmers (choice makers).

Because of the aforementioned reasons, we set two levels for coat color i.e. the most preferred is “brown” and less preferred is “white”. 

Line 182, replace the word “pictorial” with “picture”

Response: we replaced the word “pictorial” with “pictures” in line 182

Line 183, What be well educated enumerators?

Responses: the authors thank the reviewer for pointing out expressions that look vague. We replaced the phrase “well-educated enumerators” by “Enumerators with a Bachelor’s Science Degree in Animal Science …” in line 183. 

Line 186, insert article “the” before the word “profile”

Response: replaced as suggested 

Line 188, Figure #2 does not represent the two levels of choice

Reponses: the authors thank the reviewer for this important issue. We intended to show one example of the choice cards that contain one level of each attribute. Following the reviewer’s comment, we noted that the figure should contain one full choice set to show all levels of the attributes.

In our study, the number of choices in one choice set (block) is 5 (4 profiles plus 1 opt-out). Overall, we have 4 choice sets that contain 16 profiles and 4 opt-outs. This is the design we used. Our assumption to use 4 attribute profiles and opt-out was that participants can have more choice options than providing only two profile choices, and thus we can get a better understanding of which trait(level) is very important for the choice makers. Because the two levels of each attribute appeared twice but with different combinations. 

We prepared and uploaded a figure containing the 4 profiles plus the opt-out option that represents one choice set

Line 252, delete the phrase “while the remaining were female farmers”

Response: we deleted the phrase in line 252 “while the remaining were female” and corrected the sentence accordingly. 

Line 258, it is important to define what means spiritual educational system

Response: We replaced “spiritual education system” by religious schools. For the sake of ethics, we are not going to mention the type of religion but farmers were able to read and write because they were educated in such schools.

sub-topic “sheep flock size and composition” should come first than sub-topic “sheep production objective

Response: thank you for pointing out such inconsistencies, we rearranged the two sub-topics and their contents as suggested 

Line 280, replace “a” with “the”

Response: replaced as suggested 

Line 284, replace “using” with “used”

Response: replaced as suggested but we removed “been” to correct the grammar

Line 265, insert “other”

Response: inserted as suggested 

This information about the age of ram and ewe lambs and lambs must be in footnote of the table. Remember that a table must have information independent of the text.

Responses: we placed the information about the age of ram and ewe lambs and lambs as a footnote below Table 3 (line275-276) and rewritten the sentences as follow in line 269-271:

“Within the flock, ewe and ram lambs are known to be used as replacements for breeding flock or sold for income generation or otherwise slaughtered for consumption. Lambs account for about 26.6% of the total sheep flock.”

Line 330, delete “both sexes given”

Response: we deleted as suggested and corrected the sentence by inserting “who” in line 331

Line 362, the value of scale parameter (τ) is 0.7565 based on the table not 0.7535

Response: Thanks a lot, the value of the scale parameter (τ) is 0.7565, we accepted the correction

Line 367-368, delete the first sentence

Response: deleted as suggested and the second sentence (line 367-368) was modified as follow:

“The odds of selecting a ram with good tail type is 2.46 times higher than a ram with bad tail type (Table 7).”

Line 509, What would your suggestion for this?

Response: The authors thank the reviewer for raising this point. We incorporated our suggestion in line 509-510: “For example, in addition to estimated breeding values, considering coat color and tail type as selection decision criteria.” 

We know that coat color and tail type are qualitative traits that cannot be measured to estimate breeding value (unlike growth traits), this could only be addressed if selection decision is based on both subjective criteria and estimated breeding values. This, however, is at the expense of genetic gain but sustainability and acceptability of a breeding program at smallholder is expected if their need is addressed properly. Many scholars argued that the lack of success for many livestock breeding programs in developing countries is largely associated with the failure to account for the interest of smallholder farmers (Gizaw et al., 2011; Wurzinger et al., 2011; Sölkner et al., 1998).

The discussion needs more attention since repeat several times the results. I suggest review it. 

Responses: the authors thank the reviewer for raising the very important points and agreed to revise the discussion part based on the suggestion. 

We re-examined the discussion part and minimized the mentioning of the results without impairing the ideas/concepts of the remaining paragraph/sentences.

For example:

Under the sub-topic “sheep breeding practices”: 

Line 417-418, we removed the following sentence: “Next to body size and coat color, tail type and ear size were the third and fourth selection criteria for breeding rams.” And line 419-421 was rewritten. 

Line 421-422, we removed the following sentence: “For breeding ewes, female reproductive traits such as lambing interval, age at first lambing and litter size were the selection criteria considered following body size and coat color.” 

Line 427-432, we rewrote the following sentence: “In the study areas, the average production lifetime of breeding ewes is about 9.55 years implying that smallholder farmers often keep preferred ewes for a longer period. Breeding rams, on the other hand, stay within the flock relatively for short period, which is around 2.26 years on average indicating that rams provide breeding services for about one year considering a year of sexual maturity age” added in line 433-434 as “In the study areas, smallholder farmers often keep breeding ewes for longer periods while breeding rams stay within the flock relatively for a short period.” 

Furthermore, line 435 -437 was rewritten as “Yet, a large number of the smallholder farmers used rams born in the flock that could result in mating between genetically related sheep thereby increase inbreeding.”

under the sub-topic “choice preference for sheep traits”

Line 475-477, we deleted the following text:” Besides good mothering ability, predominantly brown coat color, large body size, short lambing interval, and rapid growth rate were the preferred attributes of breeding ewes by smallholder farmers” 

And line 477-479 was rewritten as “Although the tail type and litter size of ewes had shown significant influences on the preference of smallholder farmers, both were less important compared to other ewe attributes.”

Line 487-488, we deleted the sentence “However, taste heterogeneity in both sexes was not statistically significant.”

Reviewer #2

Comments:

The manuscript was well-written. Topic is very unique. I suggest to publish the manusript after minor revision.

Introduction

-I would suggest shortening the introduction in term of general context, and including more information about Breeding practices and breeding program for indigenous sheep in Ethiopia

- I would suggest describing what kind of research work/studies has been done with various; methodological approaches are available to identify breeding objective traits in sheep in Ethiopia.

Responses: the authors thank the reviewer for the insightful comments and suggestions. 

We have included the following texts (line 60-70) to describe the sheep genetic improvement efforts attempted so far in Ethiopia, though such efforts address only a few indigenous sheep breeds.

“The causes for low performances of indigenous sheep are known to be multilateral but largely related to the lack of effective breeding programs. For decades, crossbreeding between exotic sheep (such as Awassi from Israel and Dorper from South Africa) and some indigenous sheep breeds have been performed [8, 9]. However, achievements are far below expectations due to the lack of effective crossbreeding strategies and poor adaptability of crossbreeds having higher exotic genotypes [8]. Recently, community-based sheep breeding programs have been designed and implemented for Menz, Afar, Bonga and Horro indigenous sheep breeds [10, 11]. Nevertheless, the majority of the indigenous sheep breeds in different parts of the country are still managed in the traditional breeding system without being supported by proven scientific methodologies and the state of art technologies in animal breeding.” 

We have excluded the following texts (line 70-77) which seems general expression.

“It has been known that indigenous sheep are not selected systematically for a specific purpose but instead, developed through natural selection and raised for millennia under the traditional management system. Through such episodes, indigenous sheep are assumed to be merited largely with adaptive traits that are useful to survive and reproduce in low-input and stressful production environments [8]. The problem is that the traditional breeding system implemented at the smallholder level has not been supported with proven scientific methodologies and the state of art technologies in animal breeding. As a result, indigenous sheep have been performing poorly year in and year out, which is a common scenario in the northwest highlands.”

Furthermore, we modified line 85-93 to describe the methodologies that have been used to identify sheep breeding objective traits for different sheep breeds in Ethiopia and chosen the methods that fit our objective.

“Various methodological approaches have been used to identify breeding objective traits in sheep in Ethiopia. For instance, choice experiments [12], ranking among a list of traits [13, 14], and live animal rankings [10, 15]. Recently, a choice experiment has been widely applied to investigate farmers’ preferences for animal traits. It provides a hypothetical depiction of attributes levels giving adequate options for the respondents to reflect their interests. Although it requires higher cognitive efforts, a choice experiment is useful to identify preferences when the number of sheep per household is very small to conduct live animal ranking method [16]. Overall, applying a combination of methods has been suggested to effectively identify breeding objective traits [17].”

Materials and methods

- The survey area: It is good to have the map but also add some statistics. How many square miles, relative size to the whole of Ethiopia, what is the population of people, livestock species in the region of study versus the whole country etc.

Responses: we thank the reviewer for the comments.

We included the following table (Table 1) that contains information about area coverage, human population, cattle, sheep and goat population based on the administrative hierarchy (Ethiopia, Amhara region, South Gondar Zone and the study districts). This was described in the text from line 113 -115.

Table 1. Description of the study districts in comparison to the national, regional and zonal level.

 Ethiopia c Amhara Region c South Gondar Zone c Study districts d

 Farta Lay Gayient Estie

Area a 426,400 59,733.46 5442.18 424.42 587.81 527.1

Human b 94,351,001 21,134,988 2,484,929 272,177 251,926 251,708

Cattle 60,392,019 16,148,390 1,808,185 213,188 120,579 190,853

Sheep 31,302,357 11,086,083 1,085,652 113,978 88,836 191,985

Goat 32,738,385 7,766,661 514,746 51,556 48,758 104,604

aarea coverage is in square miles (Source: [18])

bhuman population projection of Ethiopia for 2017 (|Source: [19])

ccattle, sheep and goat populations of Ethiopia, Amhara Region and South Gondar (Source: [1]

d cattle, sheep and goat populations of the three districts (Source: South Gondar Zone Livestock Department annual report for 2017, Unpublished)

- How were the sampled villages/ districts chosen? Give more details. What information was there before hand? Explain further, how you conducted your focus group discussions.

Responses: the authors appreciate the comments and addressed them in the following ways.

In the sub-topic “description of the study areas”, we mentioned (modified the previous texts) why the three districts were selected. 

For example, have huge potentials for sheep production compared to other eight districts of the zone, for instance, in terms of the availability of communal grazing land and large sheep population (line 106-108)

Furthermore, the three districts are known to be the main pure breed sources for one of the sheep breeds we intend to design breeding programs, thus incorporating these areas in the study was assumed to be essential (line 108-110 previous text). 

We included a sub-topic “sample areas and focus-group discussion” (from line 127-140) to explain how the sample areas and focus-group participants were selected, how the discussion was conducted in each of the three districts. 

“Sample areas and focus-group discussion 

Secondary data about livestock population and distribution and availability of infrastructures were obtained from the Agriculture Office of the respected districts. Based on sheep population size and accessibility for transport services, three kebeles (the lowest formal administrative units in the district) were selected from each district. For focus-group discussion, a list of smallholder farmers who are regarded as knowledgeable in sheep breeding was identified with the help of livestock development agents and administrative staffs of the kebele. Then, three participants per kebele were randomly selected from the list. For each district, a separate focus-group discussion was held by involving nine well-experienced farmers, a livestock expert and researcher with the latter two played facilitation roles. During the discussion, a list of ram and ewe traits was identified taking into account their socio-cultural and economic importance both at the local and national levels. To prioritize traits, each participant farmer was asked to rank the traits based on their indigenous knowledge. Finally, the ranks from the three districts were combined and analyzed to identify the most important traits of rams and ewes in which choice experiments were designed. “ 

-The quality of Figure 2 is not good. Moreover, for the method choice scenario, it is advisable to use 2 choice scenarios and a third scenario called "no choice". 

Reponses: the authors thank the reviewer for this important issue. Our intention while preparing this figure (now it is Fig1) was to show one example of the choice cards that contain one level of each attribute. Following the reviewer’s comment, we noted that the figure should contain one full choice set to show all levels of the attributes.

In our study, the number of choices in one choice set (block) is 5 (4 profiles plus 1 opt-out). Overall, we have 4 choice sets that contain 16 profiles and 4 opt-outs. This is the design we used. 

Making a choice from 2 choice scenarios (otherwise opt-out) may be relatively easy for the smallholder farmers as their literacy status is low but it may also limit their interest because of the restricted choice freedom (as to our understanding). For example, choosing one item from 4 options and 2 options do not have equal choice freedom, which might also affect the reflection of the real interest of the choice maker.

Our assumption to use 4 attribute profiles and opt-out was that participants can have more choice options than providing only two profile choices, and thus we can get a better understanding of which trait(level) is very important for the choice makers. Because the two levels of each attribute appeared twice but with different combinations. Duguma et al. (2011) used six profile plus an opt-out to identify sheep trait preference in 4 Ethiopian indigenous sheep breeds. But making the number of choices too many may also pose problems such as choice dilemma or failure to clearly understand the difference between choice options. Thus, we assumed that a choice among 4 profiles (otherwise opt-out) is fair. 

Regarding the figure, we prepared and uploaded a figure containing the 4 profiles plus the opt-out option that represents one choice set (each level for each attribute appeared twice with different combinations.

How can you further explain your approach and justify its limitations?

Responses: the authors thank the reviewer for raising such important issues. 

We applied a questionnaire survey to identify sheep breeding practices, which is a common and appropriate method for such kind of study.

Regarding the choice experiment method, it has its own merits and limitations. The methods, in which their applicability for identifying breeding objective traits in sheep in Ethiopia have been tested, are choice experiment (Duguma et al., 2011), ranking among a group of traits (Gizaw et al., 2010; Edea et al., 2012) and live sheep ranking also known as own flock ranking or phenotypic experiment (Mirkina, 2010; Gebre 2018). 

Among the methods: choice experiment and ranking of traits require a hypothetical representation of choices. Despite their hypothetical nature, these methods are very helpful for studies carried out at smallholder level, for instance in our study areas, when sheep flock size (breeding ram and ewe) per smallholder farmers is very few. Thus, when flock size is very small, using live animal ranking method to identify breeding objective traits is less appropriate. That is why we opted to use a choice experiment (for trait preference) and ranking of traits (for breeding practice).

Besides its hypothetical nature, a choice experiment has limitations (Complexity) when large number of traits and levels are studied at the same time. Thus, we used six traits for ram and seven traits for ewes with two levels each.

In addition to what has been stated previously, we have added the following text to explain and justify our research approach (line 157-163)

“Valuing of non-marketable traits is the typical usefulness of the choice experiment method in animal breeding. Furthermore, in low input production systems where smallholders’ literacy level is low and performance recording is virtually absent, trait preference could be better elucidated using choice experiment method [10, 20]. However, when a choice is made among too many tasks, a choice experiment may not reveal true preferences due to biases allied with choice complexity [21]. Thus, the number of attributes and levels should be manageable in size to minimize the complexity of the choice experiment design.”

-You can use the articles below to enrich the Materials & Methods section and the discussion section:

1. Traoré, B., Govoeyi, B., Hamadou, I. et al. Analysis of preferences of agro-pastoralists for the attributes of traction dromedaries in harness cultivation: A case study of the Koro district of Mali. Pastoralism 9, 19 (2019). https://doi.org/10.1186/s13570-019-0153-9

2. Tindano, K., Moula, N., Traoré, A. et al. Assessing the diversity of preferences of suburban smallholder sheep keepers for breeding rams in Ouagadougou, Burkina Faso. Trop Anim Health Prod 49, 1187–1193 (2017). https://doi.org/10.1007/s11250-017-1315-7

3. Siddo, S, Moula, N, Hamadou, I, Issa, M, Marichatou, H, Leroy, P and Antoine-Moussiaux, N 2015. Breeding criteria and willingness to pay for improved Azawak zebu sires in Niger. Archive of Animal Breeding 58, 251–259.

4. Issa Hamadou, Nassim Moula, Seyni Siddo, Moumouni Issa, Hamani Marichatou, Pascal Leroy, and Nicolas Antoine-Moussiaux. Valuing breeders' preferences in the conservation of the Koundoum sheep in Niger by multi-attribute analysis. https://www.arch-anim-breed.net/62/537/2019/aab-62-537-2019.html

Responses: the authors thank the reviewer for providing very useful articles. We used these articles (cited 3) to improve the methodology and discussion parts of our manuscript. 

-I suggested to include the survey questionnaire.

Responses: We have included both the English and Amharic (local language) versions of the questionnaire as Supporting information. 

Results

- The chaper was very well described.

Discussion:

- The chaper was very well described.

-The conclusion should be developed on the outlook for your work.

Responses: thank you for the nice comment about the conclusion and we noted that the text in line 517-518 may not be in the outlook of our work, thus replaced with the following texts (line 519 -520):

“Given the absence of any form of sheep performance recording system at the smallholder level, the breeding practices and decisions mainly rely on observable characteristics of sheep.”

-The paper is very interetsing, but it can still be improved by exploiting all the collected data.

Responses: we thank the reviewer for the comment. We have used all the data (secondary and primary data) collected for this work. 

References 

1. Sölkner J, Nakimbugwe H, Zárate VA. Analysis of determinants for success and failure of village breeding programs: Proceedings of the 6th World Congress on Genetics applied to livestock production, 1998 January 11-16, Armidale, New England: University of New England; 1998.

2. Wurzinger M, Sölknera J, Iniguez L. Important aspects and limitations in considering community-based breeding programs for low-input smallholder livestock systems. Small Ruminant Research, 2011; 98: 170–175.

3. Gizaw S, Getachew T, Tibbo M, Haile A, Dessie T. Congruence between selection on breeding values and farmers’ selection criteria in sheep breeding under conventional nucleus breeding schemes. Animal, 2011; 5(7): 995-1001.

4. Bimerow T, Yitayew A, Taye M, Mekuriaw S. Morphological characteristics of Farta sheep in Amhara Region, Ethiopia. Online Journal of Animal Feed Research, 2011; 1(6):299-305.

5. Haile A, Wurzinger M, Mueller J, Mirkena T, Duguma G, Mwai O, et al. Guidelines for setting up community-based sheep breeding programs in Ethiopia. ICARDA tools and guidelines No.1, International Center for Agricultural Research in the Dry Areas; 2011

6. Duguma G, Mirkena T, Haile A, Okeyo AM, Tibbo M, Rischkowsky B, et al. Identification of smallholder farmers and pastoralists’ preferences for sheep breeding traits: a choice model approach. Animal, 2011; 5(12):1984–1992.

7. Gizaw S, Komen H, van Arendonk JAM. Participatory definition of breeding objectives and selection indexes for sheep breeding in traditional systems. Livestock Science, 2010; 128:67–74.

8. Edea Z, Haile A, Tibbo M, Sharma AK, Sölkner J, Wurzinger M. Sheep production systems and breeding practices of smallholders in western and south-western Ethiopia: Implications for designing community-based breeding strategies. Livestock Research for Rural Development, 2012; 24 (7).

9. Mirkena M. Identifying Breeding Objectives of Smallholders/Pastoralists and Optimizing Community-Based Breeding Programs for Adapted Sheep Breeds in Ethiopia. PhD Thesis, University of Natural Resources and Life Sciences, Vienna, Austria. 2010.

10. Gebre KT, Yfter KA Teweldemedhn TG, Gebremariam T. Production objectives, selection criteria and breeding practices of afar sheep in Abaala, afar region, Ethiopia. Journal of the drylands, 2018; 8(2): 834-845.

---

## [Decision Letter · Decision Letter 1]

20 Apr 2020

PONE-D-20-02104R1

Breeding practices and trait preferences of smallholder farmers for indigenous sheep in the northwest highlands of Ethiopia: Inputs to design a breeding program

PLOS ONE

Dear Dr Abebe

Thank you for submitting your manuscript to PLOS ONE. After careful consideration, your manuscript requires some very minor modifications prior to acceptance. Therefore, we invite you to submit a revised version of the manuscript that addresses the points raised during the review process.

Many thanks for resubmitting your manuscript to PLOS One

Your manuscript was reviewed by the expert reviewers and they are happy to accept this subject to a proof read

To aid with this, I have performed the proof read of the manuscript for you and made some minor suggestions.

If you can make these minor modifications, the manuscript will be accepted. Please dont feel that you need to write a response to reviewers comments.

I wish you the best of luck with the minor modifications

Hope you are keeping safe and well in this difficult time

Thanks

Simon

We would appreciate receiving your revised manuscript by Jun 04 2020 11:59PM. To enhance the reproducibility of your results, we recommend that if applicable you deposit your laboratory protocols in protocols.io, where a protocol can be assigned its own identifier (DOI) such that it can be cited independently in the future. For instructions see: http://journals.plos.org/plosone/s/submission-guidelines#loc-laboratory-protocols

A marked-up copy of your manuscript that highlights changes made to the original version. This file should be uploaded as separate file and labeled 'Revised Manuscript with Track Changes'.An unmarked version of your revised paper without tracked changes. This file should be uploaded as separate file and labeled 'Manuscript'.

We look forward to receiving your revised manuscript.

Kind regards,

Simon Russell Clegg, PhD

Academic Editor

PLOS ONE

Reviewers' comments:

Reviewer's Responses to Questions

**Comments to the Author**

1. If the authors have adequately addressed your comments raised in a previous round of review and you feel that this manuscript is now acceptable for publication, you may indicate that here to bypass the “Comments to the Author” section, enter your conflict of interest statement in the “Confidential to Editor” section, and submit your "Accept" recommendation.

Reviewer #1: All comments have been addressed

Reviewer #3: (No Response)

2. Is the manuscript technically sound, and do the data support the conclusions?

Reviewer #1: Yes

Reviewer #3: Yes

3. Has the statistical analysis been performed appropriately and rigorously? 

Reviewer #1: Yes

Reviewer #3: Yes

4. Have the authors made all data underlying the findings in their manuscript fully available?

Reviewer #1: Yes

Reviewer #3: Yes

5. Is the manuscript presented in an intelligible fashion and written in standard English?

Reviewer #1: Yes

Reviewer #3: Yes

6. Review Comments to the Author

Reviewer #1: (No Response)

Reviewer #3: As the expert reviewers recommended that the manuscript be reviewed by a native English speaker, I have carried out this review for you. The comments below are generally minor. If you could make the suggested changes (mainly just additions of commas), the manuscript can then be accepted. Please don’t write a detailed reviewer response, just a simple line saying that they were all done is more than sufficient.

Line 27- …sheep attributes, while descriptive ….

Line 29- …keeping sheep, followed by …

Line 33- …selection criteria, followed by …

Line 40- change ‘objective’ with ‘objectives’

Line 51- …of the country, where Farta sheep ….

Line 58- …for Farta sheep, while Gullilat ….

Line 61- …multilaterally, but largely related …

Line 65- ..crossbreeds, due to having higher exotic genotypes (I think that was what you were trying to say but please check)

Line 68- ..breeding system, without being …

Line 73- … have not yet been identified

Line 77- sheep in Ethiopia, for instance … (link the two sentences)

Line 79- change attributes to attribute

Line 80- …attribute levels, giving adequate ….

Line 82- …conduct a live animal ranking method …

Line 85- highlands of Ethiopia, using a ranking …

Line 93- should recoded be recorded?

Line 97 …production, compared to the other eight ….

Line 102- …mixed farming, where livestock ….

Line 102- …and the livelihood of small ….

Line 108- …average daily temperatures, whereas the minimum …. (link the two sentences )

Line 124- … and researcher, with the latter …

Line 126- ..importance, both at the ….

Line 132- ….interviews, using a ….

Line 133- …[16] , who ….

Line 167- …picture representations ….

Line 180- …alternatives, excluding …

Line 220- …setting an appropriate initial value….

Line 227- …preferences, thus were omitted ….

Line 243- …school, while ….

Line 251- delete ‘is’. (…of breeding ewes accounted for …)

Line 269- …controlling external parasites ….

Line 275- …breeding rams, but all …

Line 276- …had access to ram services ….

Line 277- replace ‘staying’ with ‘stayed’

Line 286- …breeding rams, followed by …

Line 302- …was 9.78 years, while rams were ….

Line 319- ….presence of horns ….

Line 320- ….preferred over a ram with small ears.

Line 321- ..tail type of the ram is the most …

Line 321- …preferred attribute, while ear size …

Line 324- … preferred ewe attribute, followed by predominantly

Line 326- …preferred attributes, while good tail ….

Line 326- is this twining or twinning (have 2 offspring)?

Line 340- …levels, ewes with good ….

Line 342- … than ewes with poor…

Line 362- ..in a crop-livestock ….

Line 362- …system, while ….

Line 372- …qualitative nature, implying that ….

Line 384- …characterized by a fatty tail type ….

Line 385- …orientated ears ….

Line 391- …longer periods, while breeding rams ….

Line 392- stay within the flock for a relatively short period. (reword)

Line 394- …in the flock, that could ….

Line 395- …related sheep, thereby increasing ….

Line 404- …selection criteria, although the ..

Line 420- Unexpectedly, a ram with small ear size was preferred to a ram with larger ears, although ear size is the least important attribute in terms of the magnitude of the utility coefficient.

Line 429- …to be profitable, even under ….

Line 429- One reason could be…. (reword)

Line 440- Unlike for the breeding ram, ewe preference analysis revealed significant scale heterogeneity …..

Line 443- …one or a few attributes….

Line 452- change ‘objective’ to ‘objectives’

Line 453- …for body size, while growth can …

Line 456- …ability of ewes that highly influenced …

Line 459- …thus are difficult to incorporate ….

Line 459- …breeding objective, despite their …

Line 460- change ‘decision’ to ‘decisions’

Line 462- …should be taken into account, for example, in addition to … (merge sentences into one)

Line 463- remove because. (It has been said that…)

7. PLOS authors have the option to publish the peer review history of their article (what does this mean?). If published, this will include your full peer review and any attached files.

Reviewer #1: No

Reviewer #3: Yes: Simon Clegg

---

## [Author Response · Author response to Decision Letter 1]

25 Apr 2020

Dear PLOS ONE Academic Editor, thank you for the comments. We addressed all the comments based on the suggestion.

---

## [Editor Report · Decision Letter 2]

28 Apr 2020

Breeding practices and trait preferences of smallholder farmers for indigenous sheep in the northwest highlands of Ethiopia: Inputs to design a breeding program

PONE-D-20-02104R2

Dear Dr. Abebe

We are pleased to inform you that your manuscript has been judged scientifically suitable for publication and will be formally accepted for publication once it complies with all outstanding technical requirements.

With kind regards,

Simon Russell Clegg, PhD

Academic Editor

PLOS ONE

Additional Editor Comments (optional):

Many thanks for resubmitting your manuscript to PLOS One

I have reviewed your manuscript and wish to thank you for addressing the reviewer comments.

I have recommended your manuscript for publication, and you should hear from the Editorial Office soon

It was a pleasure working with you, and I wish you all the best for your future research

Hope you are keeping safe and well in these difficult times

Thanks

Simon

---

## [Editor Report · Acceptance letter]

30 Apr 2020

PONE-D-20-02104R2 

Breeding practices and trait preferences of smallholder farmers for indigenous sheep in the northwest highlands of Ethiopia: Inputs to design a breeding program 

Dear Dr. Abebe:

I am pleased to inform you that your manuscript has been deemed suitable for publication in PLOS ONE. Congratulations! Your manuscript is now with our production department. 

With kind regards,

on behalf of

Dr. Simon Russell Clegg 

Academic Editor

PLOS ONE